# Models and Interpretation Methods for Single-Hole Flowmeter Experiments

Gerard Lods and Delphine Roubinet *

Geosciences Montpellier, Montpellier University, CNRS, 34090 Montpellier, France; gerard.lods@umontpellier.fr
* Correspondence: delphine.roubinet@umontpellier.fr

**Abstract:** Subsurface and groundwater flow characterization is of great importance for various environmental applications, such as the dispersion of contaminants and their remediation. For single-hole flowmeter measurements, key characteristics, such as wellbore storage, skin factor heterogeneities, and variable pumping and aquifer flow rates, have a strong impact on the system characterization, whereas they are not fully considered in existing models and interpretation methods. In this study, we develop a new semi-analytical solution that considers all these characteristics in a physics-based consistent manner. We also present two new interpretation methods, the Double Flowmeter Test with Transient Flow rate (DFTTF) and the Transient Flow rate Flowmeter Test (TFFT), for interpreting data collected during single and multiple pumping tests, respectively. These solution and methods are used as follows. (i) The impact of wellbore storage, transient pumping rate, and property heterogeneities on the interpretation of data collected during single pumping tests are studied over 49 two-aquifer cases. (ii) The effect of the skin factor heterogeneity on transmissivity and storativity estimates, as well as the variability range of the (non-unique) corresponding solutions, are analyzed for the interpretation of multiple-pumping experiments. The results presented in this work show the importance of the various properties and processes that are considered, and the need for the new models and methods that are provided.

**Keywords:** flowmeter experiments; semi-analytical solutions; hydraulic property estimates; interpretation methods; analysis and inversion of transient behavior

## 1. Introduction

Characterizing subsurface and groundwater flow is performed by collecting data from several techniques, including hydraulic tests, thermal experiments and electrical measurements (e.g., [1–4]), and inverting this data with the most appropriate inversion strategies (e.g., [5–8]). Among all these methods, the most-used method is the pumping test, which gives global estimates of the hydraulic properties of the system and provide information on its transient behavior when interpreting the data with transient-flow solutions (e.g., [9–12]). When characterizing the vertical heterogeneities of the system is required, for investigating the dispersion of contaminant and planning remediation strategies for instance, additional information can be acquired by downhole well logging measurements including temperature, vertical flow rates, and direct observations with optical and acoustic imaging tools (e.g., [13–15]).

Using heat pulse and electromagnetic tools [13,16–18] results in measuring vertical flow rates that are as small as 0.05 Lpm. These measurements are used either in a qualitative way to identify the most permeable zones of the system, or in a quantitative manner to evaluate the hydraulic properties of these zones [19–21]. In the latter case, the properties are estimated from vertical flow rate measurements that are collected above each conductive zone in single- or cross-borehole configurations (e.g., [14,22,23]). Two main methods are used to collect this data. (i) Log measurements are acquired during a single pumping test, which is easy and quick to perform, and results in collecting one or two values of flow rates

above each conductive zone while the logging tool is lowered in the well (e.g., [1,24,25]). (ii) Multiple pumping tests are performed with the logging tool being located above a different conductive zone for each test, resulting in monitoring the changes in vertical flow rates over time (e.g., [14,26,27]). Method (ii) corresponds to series of local experiments that provide more information than method (i), while being time consuming, since it requires to conduct more experiments and to wait until that the system goes back to equilibrium between each experiment.

The corresponding data are usually interpreted with standard analytical solutions that represent the identified conductive zones as independent aquifers, whose behavior is described with the Theis Equation [28–30]. Alternatively, double-layer numerical models are used to consider more complex configurations, including vertical crossflow between adjacent porous layers or skin effect with a homogeneous skin zone located around the well [31–33]. Heterogeneous skin factors, wellbore storage, and transient pumping rate are considered separately in different studies such as the semi-analytical multi-aquifer models developed in [34,35] and the methods formulated in [14,26]. These three features describe phenomena occurring in the pumping well and its vicinity that are important for the interpretation of pumping tests. (i) The skin factor enables one to take into account the skin effects that are related to headloss in the well and in its vicinity in the context of significant differences between the well vicinity and aquifer hydraulic conductivities. These differences are due to, for instance, mud invasion or turbulence in the well and its vicinity [9]. (ii) The wellbore storage impacts the data collected at the beginning of the experiment and the solutions that neglect this characteristic must be applied when the wellbore storage effect is completed. This requires to conduct long-time experiments, in particular for low-permeability aquifers with positive skin effects and large well diameters [30,36]. (iii) In addition to intentional changes related to the experiment need, changes in pumping flow rate along pumping experiments are very frequent due to the pump functioning, with a decrease in flow rate when the drawdown increases. These changes can also be due to potential clogging of the pump or the command valve, and must be accounted for when the decrease is important [27].

Some of the solutions cited above rely on strong assumptions regarding the vertical heterogeneities of the considered systems, assuming for instance homogeneous hydraulic diffusivities and storativities in the system. Furthermore, none of these solutions considers at the same time the effect of wellbore storage, skin factor heterogeneities, and variable pumping and aquifer flow rates, whereas the importance of each of those characteristics has been demonstrated separately. In this study, we develop a new semi-analytical solution that considers all these key characteristics and that is used to generate synthetic reference data. We also remind the assumptions related to standard interpretation methods and we analyze their impact on the interpretation of single-hole flowmeter measurements that are collected over single pumping tests. For the interpretation of flowmeter data collected during single or double logs along the well, a new interpretation method is presented to improve the estimation of aquifer properties, and evaluate the impact of standard assumptions over a large range of cases. For the interpretation of flowmeter data collected during multiple pumping tests, where the full transient behavior of vertical flow rates above each conductive zone is recorded, we also present a new interpretation method that is easy to implement. This method is used to evaluate the variability range of key properties that drive the transient behavior of the system, which are the storativity and skin factor. The validity and accuracy of each model and interpretation method are discussed through synthetic cases, before providing a discussion and conclusions on this work.

## 2. Experiments and Interpretation Methods

### 2.1. Considered Experiments and Methods

The analysis of flowmeter tests provided in this study is performed by considering the following kinds of experiments (Table 1).

- Single-pumping single-log experiments. The vertical flow rates and hydraulic heads are measured along the borehole while the flowmeter is lowered into the well during a single pumping test.
- Single-pumping double-log experiments. As before, the vertical flow rates and hydraulic heads are measured along the borehole during two log experiments that are conducted under the same pumping test.
- Multiple-pumping local-log experiments. A pumping test is performed for each conductive zone that needs to be characterized (except the upper one) with the logging tool localized above this zone.

**Table 1.** Considered flowmeter experiments with the following collected data: $h_{w,i}$ and $q_i$ are the hydraulic heads and vertical flow rates, respectively, measured above conductive zone $i$ with $N_{aq}$ the number of conductive zones to characterize, and $h_{w,i}^j$ and $q_i^j$ are their counterpart collected from two logs ($j = 1, 2$). SFT, DFT, DFTTF, and TFFT are the interpretation methods presented in Appendix A and Section 2.

| Experiment Name | Collected Data | Interp. Methods |
|---|---|---|
| Single-pumping single-log | $(h_{w,i}, q_i), i = 1, \ldots, N_{aq}$ | SFT |
| Single-pumping double-log | $(h_{w,i}^j, q_i^j), j = 1, 2$ | DFT, DFTTF |
| Multiple-pumping local-log | $(h_{w,i}(t), q_i(t))$ | TFFT |

As described in Table 1, these experiments are interpreted with the methods SFT, DFT, DFTTF, and TFFT. SFT and DFT are standard methods whose basis are reminded in Appendix A, and DFTTF and TFFT correspond to new methods that are described below. For the methods presented below, the vertical flow rates $q_i$ ($i = 1, \ldots, N_{aq}$) measured above aquifer $i$ are expressed in terms of aquifer flow rates $Q_i$ ($i = 1, \ldots, N_{aq}$) with the differentiation method described in expressions (A2).

*2.2. Double Flowmeter Test with Transient Flow Rate (DFTTF)*

The Double Flowmeter Test with Transient Flow rate (DFTTF) method is based on the DFT, which consists of evaluating the transmissivity $T_i$ and storativity $S_i$ of aquifer $i$ from Theis' solution using two measurements of the vertical flow rates and hydraulic head above each conductive zone (see the description of the DFT method in Appendix A.2). While the DFT method relies on the assumption that the values of the considered vertical flow rates are equal to their average value, the DFTTF method considers these values as distinct without assumption. In the conditions of validity of the logarithmic approximation of the Theis function, the hydraulic properties of aquifer $i$ are expressed as:

$$T_i = \frac{1}{4\pi} \frac{\ln(t_{i2}/t_{i1})}{h_{wi}(t_{i2})/Q_i(t_{i2}) - h_{wi}(t_{i1})/Q_i(t_{i1})} \tag{1a}$$

$$S_i = \frac{2.25 T_i t_{i1}}{r_{wi}^2} \exp\left(\frac{2\sigma_i - 4\pi T_i(h_{wi}(t_{i1}))}{Q_i(t_{i1})}\right) \tag{1b}$$

with the aquifer flow rates $Q_i(t_{i1})$ and $Q_i(t_{i2})$ and hydraulic heads $h_{wi}(t_{i1})$ and $h_{wi}(t_{i2})$ deduced from measurements performed at times $t_{i1}$ and $t_{i2}$ for aquifer $i$. In expression (1), $r_{wi}$ and $\sigma_i$ are the well radius and skin factor associated with aquifer $i$, respectively, $\sigma_i$ being considered as homogeneous over the aquifers (i.e., $\sigma_i = \sigma$) and defined from pump test interpretation, as in DFT. Note that in expression (1b), $S_i$ is expressed as a function of $t_{i1}$ and $T_i$, but could also be expressed as a function of $t_{i2}$ and $T_i$ by replacing $t_{i1}$ with $t_{i2}$. Since $T_i$ depends on both $t_{i1}$ and $t_{i2}$, $S_i$ also depends on both times regardless of the chosen expression.

*2.3. Transient Flow Rate Flowmeter Test (TFFT)*

The TFFT (Transient Flow rate Flowmeter Test) method is developed to interpret multiple-pumping local-log experiments. These experiments consist of performing a pumping test for each aquifer to characterize (except the upper one) with the same pumping rate $Q_P$. The resulting drawdown $h_{wS}$ and the full transient behavior of the vertical flow rates $q_i$ above aquifers $i$ are recorded and the corresponding aquifer transient flow rates $Q_i$ are deduced as explained before. This leads to the set of data $(Q_i(t), Q_P(t), h_{wS}(t))$ associated with each aquifer $i$ ($i = 1, \ldots, N_{aq}$) and collected during $N_P = N_{aq} - 1$ pumping tests. These data are inverted with the following algorithm.

1. Model the data $Q_P(t)$ with variable pumping flow rate models, as described in Appendix C.1.
2. Estimate the unknowns $T_i$, $S_i$, $\sigma_i$ by inverting the data $(Q_i(t), h_{wS}(t))$ provided by the multi-aquifer model described in Appendix B using a numerical optimization method. More precisely, the Laplace transform of $Q_i(t)$ and $h_{wS}(t)$ are given in expressions (A15) and (A17) and numerically inverted with [37]'s algorithm.

The data inversion proceeds iteratively by a least square method using a gradient algorithm, applied to the transient drawdown and aquifers flow rates, for which different weights can be considered. The objective function is expressed as:

$$f_{obj} = \sum_t \alpha^t \left( \frac{h_{wS}^{model}(t) - h_{wS}^{data}(t)}{h_{wS}^{data}(t)} \right)^2 + \sum_i \sum_t \alpha_i^t \left( \frac{Q_i^{model}(t) - Q_i^{data}(t)}{Q_i^{data}(t)} \right)^2 \tag{2}$$

with the weights $\alpha^t$ and $\alpha_i^t$, and the reference $(Q_i^{data}(t), h_{wS}^{data}(t))$ and simulated $(Q_i^{model}(t), h_{wS}^{model}(t))$ data. Note that this objective function is used in Section 3.3 with synthetic data that are considered every minute and weights that are set to 1. The inversions are performed from an initial value given by the DFTTF method with null skin factors and convergence is assumed when the relative change of the objective function is less than $10^{-12}$ in the last five iterations.

## 3. Examples of Applications on Synthetic Cases

*3.1. Considered Configurations*

We consider a system of two aquifers, $T_1$ and $S_1$ being the transmissivity and storativity of aquifer 1, respectively, and $T_2$ and $S_2$ of aquifer 2. We evaluate how these properties are estimated from single- and double-log flowmeter experiments with the interpretation methods SFT, DFT, and DFTTF in various context in terms of wellbore storage and transient pumping rate (Section 3.2). Then, we evaluate these properties by taking into account different skin factors in two-aquifer systems considering wellbore storage and transient pumping rate with an analysis of the range of the storativity and skin factor that can be defined (Section 3.3). This is done by considering series of local flowmeter experiments that are interpreted with the TFFT method. For all the cases presented in this section, the reference synthetic data are provided by the multi-aquifer model that is presented in Appendix B and takes into account the wellbore storage effect, transient pumping flow rate, and heterogeneous skin factor. A homogeneous pumping well of radius 8 cm is considered with aquifers of 1 m thickness, resulting in transmissivity and storativity equal to hydraulic conductivity and specific storage, respectively.

*3.2. Single- and Double-Log Flowmeter Experiments*

The considered two-aquifer system is defined by setting $T_1$ and $S_1$ to $10^{-4}$ m$^2$/s and $10^{-3}$, respectively, and having $T_2$ and $S_2$ ranging from $10^{-7}$ to $10^{-1}$ m$^2$/s and $10^{-6}$ to 1, respectively, with null skin factor in both aquifers. The three following configurations are considered: (i) standard models without accounting for wellbore storage and considering constant pumping flow rate (*Config1*), (ii) models that account for wellbore storage and consider constant pumping flow rate (*Config2*), and (iii) no wellbore storage and transient

pumping flow rate (*Config3*). The transmissivities are estimated with the interpretation methods SFT, DFT, and DFTTF, and the storativities with DFT and DFTTF, considering the measurements at time $t_1 = 10$ min for SFT and at times $t_1$ and $t_2 = 300$ min for DFT and DFTTF. Note that the small time $t_1$ allows to investigate the influence of the wellbore storage and the transient pumping rate. The corresponding results are presented in Figure 1 and the pumping flow rate $Q_P$ that is used in *Config3* is defined in Figure 2. Note that $Q_P$ is set to the constant value 4 Lpm in *Config1* and *Config2*.

The results provided in Figure 1 are described as follows. (i) When the diffusivities of aquifers $i$ (aquifer for which the properties are estimated) and $j$ (the other aquifer) are similar (i.e., $\varepsilon_{ij} \approx 1$), the ratio of estimated to true properties ($\mathcal{T}_i$ and $\mathcal{S}_i$) are equal to 1 (or very close to 1) for all the considered configurations and interpretation methods. This shows that for homogeneous hydraulic diffusivity (i.e., $\varepsilon_i \approx \varepsilon_j$), the assumptions related to the considered interpretation methods hold, in particular the assumptions related to Theis' model. (ii) For large values of $\varepsilon_{ij}$ (i.e., $\varepsilon_{ij} >> 1$), the errors in estimating the hydraulic properties are relatively small in comparison with the errors obtained for small values of $\varepsilon_{ij}$ (i.e., $\varepsilon_{ij} << 1$). In most of the cases, we observe that the error in estimating $\mathcal{T}_i$ and $\mathcal{S}_i$ increases when $\varepsilon_{ij}$ varies from 1 to $10^3$ because the assumptions described before that hold when $\varepsilon_{ij} \approx 1$ are less and less fulfilled. This error then decreases when $\varepsilon_{ij}$ varies from $10^3$ to $10^6$, which corresponds to configurations where the diffusivity of aquifer $j$ is negligible in comparison with that of aquifer $i$ for which the properties need to be estimated. This implies that aquifer $j$ poorly contributes to the pumping and that the assumption of constant pumping related to aquifer $i$ and used in Theis' model holds. (iii) For small values of $\varepsilon_{ij}$ (i.e., $\varepsilon_{ij} << 1$), the transmissivities and storativities are overestimated and underestimated, respectively. $\varepsilon_i < \varepsilon_j$ corresponds to configurations where $\mathcal{T}_i < \mathcal{T}_j$ or $\mathcal{S}_i > \mathcal{S}_j$, whereas the methods SFT and DFT rely on the assumption of homogeneous diffusivities. Overestimating $\mathcal{T}_i$ and underestimating $\mathcal{S}_i$ result in tending to the relations $\mathcal{T}_i \approx \mathcal{T}_j$ and $\mathcal{S}_i \approx \mathcal{S}_j$, and thus, tending to verify the homogeneous diffusivity assumption. (iv) For all the estimates of $\mathcal{T}_1$ and $\mathcal{T}_2$, the results obtained with SFT and DFT are similar or slightly improved by DFT, except for the estimated value of $\mathcal{T}_1$ in *Config3*. For all the cases, the DFTTF method always provides better estimates than SFT and DFT with the ratio of estimated to true value around 1 for most of the cases and reaching for example a maximum value of 3 for the transmissivity estimates. (v) Only small differences are observed between the three considered configurations, showing that the wellbore storage and transient pumping flow rate do not have a significant impact on the estimation of the considered properties. An exception is observed when estimating the storativities $S_1$ and $S_2$ in *Config3*, for which the transient pumping flow rate results in increasing the ratio of estimated to the true value of 4 to 6 orders of magnitude with the DFT method (blue symbols in Figure 1i). This results in a significant underestimation of $\mathcal{S}_1$ and $\mathcal{S}_2$.

### 3.3. Series of Local Flowmeter Experiments

We consider now the effect of different skin factors in two-aquifer configurations with $T_1$ and $T_2$ set to $5 \times 10^{-4}$ and $10^{-5}$ m$^2$/s, $S_1$ and $S_2$ to $5 \times 10^{-4}$ and $10^{-3}$, and the skin factors $\sigma_1$ and $\sigma_2$ to 0 and 1, respectively. As before, the pumping well radius is homogeneous and equal to the wellbore storage radius $r_{wS} = 8$ cm. The applied transient pumping flow rate is the same as in *Config3* of the previous example and the resulting transient aquifer flow rates are defined from the multi-aquifer model provided in Appendix B and Figure 2. These data are interpreted with the TFFT method presented in Section 2.3 and the resulting estimated values are presented in Figure 3.

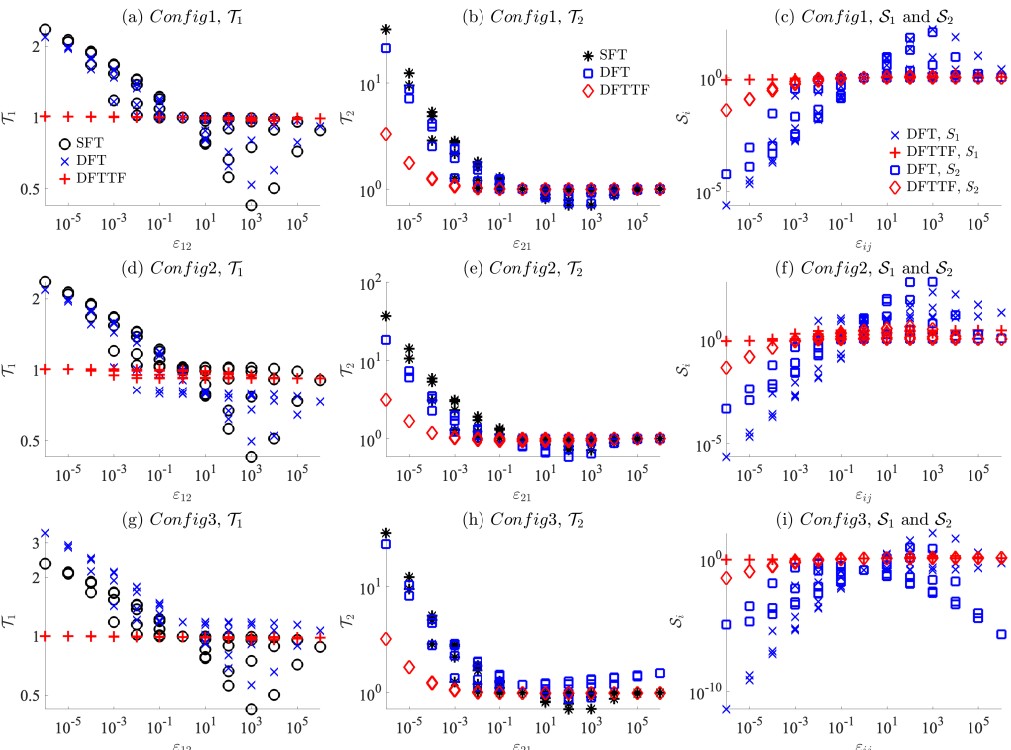

**Figure 1.** Estimated values of transmissivities and storativities normalized by the true values ($\mathcal{T}_i$ and $\mathcal{S}_i$, respectively) for aquifer 1 and 2 ($i = 1, 2$) along $\varepsilon_{ij}$ the ratio of diffusivities of aquifer $i$ to that of aquifer $j$ ($\varepsilon_{ij} = \varepsilon_i / \varepsilon_j$), for model configurations *Config1* (**first row**), *Config2* (**second row**), and *Config3* (**third row**), and the interpretation methods SFT, DFT, and DFTTF (black, blue, and red symbols, respectively).

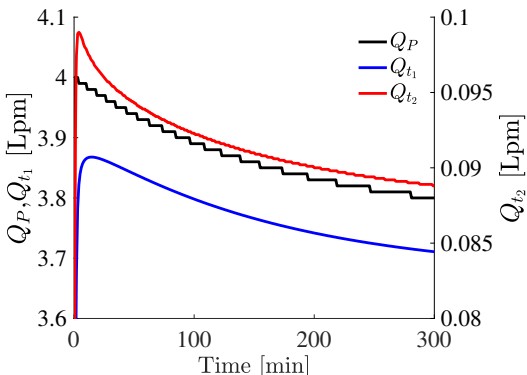

**Figure 2.** Transient pumping flow rate ($Q_P$) defined from the exponential model provided in (A18) with $Q_{t_1} = 4$ and $Q_{t_2} = 3.8$ Lpm at times $t_1 = 0$ and $t_2 = 300$ min with the fitting coefficient $b$ set to $10^5$. $Q_P$ is the pumping flow rate considered in *Config3* of Section 3.2 and in all the experiments in Section 3.3. $Q_{t_1}$ and $Q_{t_2}$ correspond to the aquifer flow rates of the reference experiments considered in Section 3.3.

Figure 3a shows the minimum value of the objective function (2) obtained for each value of couple ($\sigma_1, \sigma_2$). These results show that the smallest value is observed for the true values of $\sigma_1$ and $\sigma_2$ (i.e., $\sigma_1 = 0$ and $\sigma_2 = 1$), demonstrating that the studied objective function is well defined. The corresponding estimated values of $T_1$, $T_2$, $S_1$, and $S_2$ provided in Figure 3b–e (for $\sigma_1 = 1$ and $\sigma_2 = 0$) are equal to the true values of these properties, showing the ability of the proposed interpretation method to estimate the hydraulic properties of the considered system.

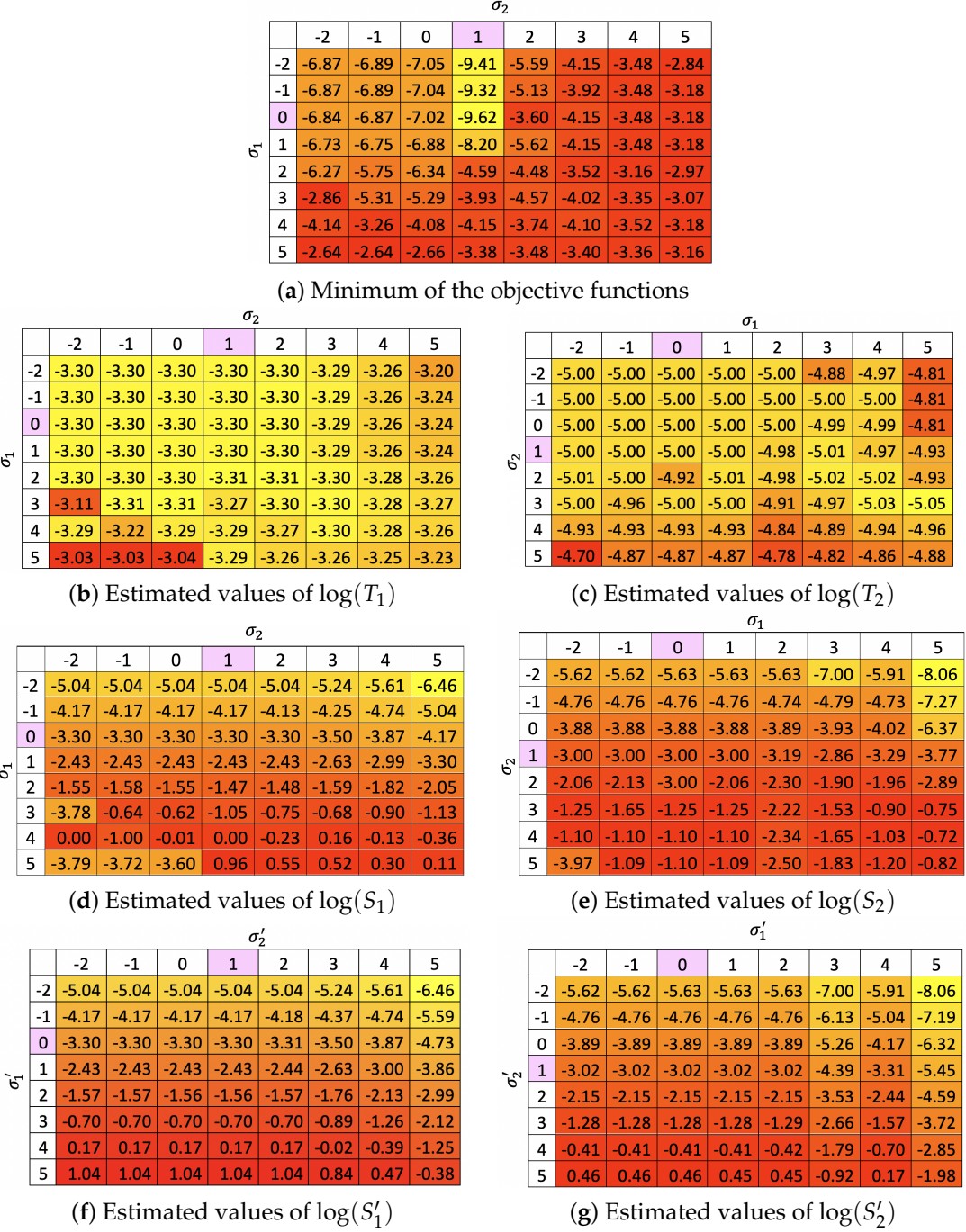

**Figure 3.** (**a**) Minimum of the objective function (2) obtained for each value of the couple $(\sigma_1, \sigma_2)$ for $\sigma_1$ and $\sigma_2$ ranging from $-2$ to 5. The corresponding estimated values of (**b**) $\log(T_1)$, (**c**) $\log(T_2)$, (**d**) $\log(S_1)$, and (**e**) $\log(S_2)$ obtained with the TFFT method. (**f**,**g**) Estimated values of $\log(S'_1)$ and $\log(S'_2)$ obtained with expression (A23) from the values of $S_1$ when $\sigma_1 = -2$ (first line in (**d**)) and $S_2$ when $\sigma_2 = -2$ (first line in (**e**)). For all figures, the color cells correspond to increasing values from yellow to red and the log function corresponds to log10 function.

The results presented in Figure 3b–e also lead to the following observations. (i) The transmissivities $T_1$ and $T_2$ are poorly affected by the values of the skin factors $\sigma_1$ and $\sigma_2$, since $T_1$ ranges from $4.92 \times 10^{-4}$ to $9.30 \times 10^{-4}$ m$^2$/s (Figure 3b) and $T_2$ from $8.97 \times 10^{-6}$ to $1.99 \times 10^{-5}$ m$^2$/s (Figure 3c). (ii) On the contrary, the storativities values $S_1$ and $S_2$ are strongly impacted by the values of $\sigma_1$ and $\sigma_2$, since $S_1$ ranges from $3.47 \times 10^{-7}$ to 9.12 (Figure 3d) and $S_2$ ranges from $8.71 \times 10^{-9}$ to $8.13 \times 10^{-2}$ (Figure 3e). (iii) The values

observed in Figure 3d,e show minimum localized values of both $S_1$ and $S_2$ for high contrast between $\sigma_1$ and $\sigma_2$. This corresponds to configurations where $\sigma_1$ is large and $\sigma_2$ small, or $\sigma_1$ small and $\sigma_2$ large, which are represented in the top right and bottom left corners of Figure 3d,e (yellow and orange cells). (iv) We also observe that high values of $\sigma_1$ and $\sigma_2$ result in overestimating $S_1$ and $S_2$ with red cells located at the bottom right corners of Figure 3d,e. High values of $\sigma_i$ correspond to high and low values of the headlosses in the skin and aquifer, respectively, corresponding to a low contribution of the aquifer to the drawdown. Trying to fit reference data that are obtained with smaller values of $\sigma_i$ results in overestimating the storativities to counterbalance the impact of large values of $\sigma_i$.

The storativity is usually determined from the drawdown recorded in a distant observation well. Determining this property from data collected in the pumping well is a challenge, since an infinity of couples $(S, \sigma)$ can reproduce the recorded drawdown, as demonstrated in Appendix C.2. The resulting relationship (A23) between $S_i$ and $\sigma_i$ is used to estimate an equivalent couple of parameters $(S_i', \sigma_i')$ that are presented in Figure 3f,g. The global behavior of $S_1$ and $S_2$ is well reproduced, except for the localized minimum values that are observed at the bottom left corners in Figure 3d,e, and not in Figure 3f,g. These differences are due to relationship (A23), which only depends on $(S_i, \sigma_i)$ when estimating $(S_i', \sigma_i')$, implying that this relationship does not consider the impact of one aquifer on the other. These results show that the localized minimum values observed in Figure 3d,e are due to the impact of the properties of aquifer $j$ when estimating the storativity of aquifer $i$.

## 4. Discussion

We present a new semi-analytical multi-aquifer model relying on an independent aquifers representation. This physics-based solution is formulated for heterogeneous aquifer properties, skin factor, and pumping well radius, taking into account the wellbore storage and various transient pumping rates and transient aquifer flow rates. We also present two new interpretation methods that (i) improve the transmissivity and storativity estimates of multi-aquifer systems, while taking into account wellbore storage and transient pumping flow rates, and (ii) help to conduct sensitivity analysis of these estimates in relation to the skin factor.

Our study shows that the interpretation of single- and double-log flowmeter experiments collected during a single pumping test is improved by using the DFTTF method. By accounting for transient aquifer flow rates, this method gives better estimates of the transmissivity and storativities of aquifers than the standard SFT and DFT methods. We also show that the interpretation of multiple-pumping local-log experiments conducted with the TFFT method leads to consistent results. It demonstrates radically different sensitivity of the transmissivities and storativities to the skin factors considered in the aquifers. A relationship between storativity and skin factor is provided and tested to analyze the (non-unique) couples of solutions related to these parameters in the context of single-hole data, while the storativity is usually estimated from data collected in an observation borehole.

However, the presented solution and methods have been only applied to two-aquifer systems and synthetic data with relatively simple configurations. Additional work is required to demonstrate the efficiency of this solution and methods on complex configurations and their ability to interpret field data. Extending the sensitivity analysis also presented in this work to complex configurations and field data is required as well to fully demonstrate the interest of this work.

## 5. Conclusions

For future work and applications, this easy-to-implement solution can be extended to account for inwell headlosses, and for interpreting the recovery phase of flowmeter experiments by applying the superposition principle. We will also consider structural information that helps to reduce the range of acceptable skin factors and associated storativities. For example, small and large storativities are unlikely in semi-confined systems and in weakly fissured confined hard rocks, respectively. Using logs of acoustic or optical

borehole images helps to verify the presence of fractures, which gives information on the skin factor. The latter is negative when the well is intersected by local well-open conducts (such as well-open fractures), small with low flow rates, null when considering fractures with homogeneous aperture and rugosity in laminar regime, and positive for fractures filled by drilling mud or sediments. The relevance of the skin factor value, which corresponds to a singular headloss at the infinitesimal interface between the well and the aquifer, can also be checked with equivalent finite skins, such as darcian skins, or laminar-turbulent fracture skins [38].

**Author Contributions:** Conceptualization, G.L. and D.R.; methodology, G.L.; software, G.L.; validation, G.L.; writing, G.L. and D.R.; review and editing, D.R.; visualization, D.R.; supervision, D.R. All authors have read and agreed to the published version of the manuscript.

**Funding:** This research received no external funding.

**Data Availability Statement:** The data that support the findings of this study are available from the corresponding author upon request.

**Conflicts of Interest:** The authors declare no conflict of interest.

## Appendix A. Standard Models and Interpretation Methods

Considering a homogeneous and infinite aquifer subject to a constant pumping flow rate $Q$ with negligible wellbore storage, ref. [39]'s solution provides the following expression of the well drawdown:

$$h_w(t) = \frac{Q}{4\pi Kb}\left[W\left(\frac{S_s r_w^2}{4Kt}\right) + 2\sigma\right] \tag{A1}$$

where $K$, $S_s$, and $b$ are the aquifer hydraulic conductivity, specific storativity, and thickness, respectively, $\sigma$ and $r_w$ are the skin factor and pumping well radius, and $W$ is the Theis function. In standard pumping tests, drawdown data are collected in a distant observation well and expression (A1) is used to estimate global values of the aquifer transmissivity and storativity that are defined as $T = Kb$ and $S = S_s b$.

In a system composed of horizontal and independent aquifers, where $T_i$ and $S_i$ are the transmissivity and storativity of aquifer $i$, single-hole flowmeter tests rely on logs that are performed along the pumping well and provide the vertical flow rates $q_i$ measured above each conductive zone $i$. The aquifer flow rate $Q_i$ of aquifer $i$, which is defined as the flow rate provided by the aquifer during the pumping experiment, is deduced from the measurements $q_i$ with the following vertical differentiation:

$$Q_i = q_i - q_{i+1}, \quad i = 1, \ldots, N_{aq} - 1,$$
$$Q_{N_{aq}} = q_{N_{aq}}, \tag{A2}$$

where $N_{aq}$ is the number of horizontal aquifers numbered from top to bottom. Estimates of the hydraulic properties of each aquifer $i$ are provided by the SFT and DFT interpretation methods for single and double flowmeter tests, each of those methods relying on different assumptions that are described below.

### Appendix A.1. Single Flowmeter Test (SFT)

The SFT method consists of interpreting aquifer flow rates $Q_i$ deduced from a single-log of vertical flow rates $q_i$ and associated with the hydraulic head $h_{w,i}$ for each conductive zone $i$. The hydraulic transmissivity $T_i$ is estimated at a given time using the following assumptions: (H1) the hydraulic diffusivities $\varepsilon_i = K_i/S_i$ are homogeneous (i.e., $\varepsilon_i = \varepsilon$, $\forall i$) [40] and (H2) the specific storativities $S_{si}$ are homogeneous (i.e., $S_{si} = S_s$, $\forall i$) [28]. Homogeneous skin factor is also usually assumed (i.e., $\sigma_i = \sigma$) and $T_i$ is inverted from (A1) for each aquifer $i$, analytically under assumption (H1) and numerically under (H2). In our study, we consider the SFT method with assumption (H1). Alternatively, considering in addition

negligible inwell headlosses (i.e., $h_{wi}(t) = h_w(t)$) and homogeneous pumping well radius (i.e., $r_{wi} = r_w$), $T_i$ is given by the well-know formula $T_i = TQ_i/Q$. This expression has been deduced from numerical simulations in [29] for horizontal layered systems with cross-flow exchanges under pseudo-steady state conditions, and from analytical demonstration in [30], as done here for independent layered systems under transient conditions.

*Appendix A.2. Double Flowmeter Test (DFT)*

Alternatively, the DFT method relies on two logs of data which provide the aquifer flow rates $Q_i(t_{i1})$ and $Q_i(t_{i2})$ at two distinct times $t_{i1}$ and $t_{i2}$. Assuming homogeneous skin factors as before, and being in the conditions of validity of the logarithmic approximation of the Theis function [39], the following explicit expressions of $T_i$ and $S_i$ are obtained: [30]

$$T_i = \frac{\tilde{Q}_i}{4\pi(h_{wi}(t_{i2}) - h_{wi}(t_{i1}))} \ln\left(\frac{t_{i2}}{t_{i1}}\right) \tag{A3a}$$

$$S_i = \frac{2.25 T_i t_{i1}}{r_{we}^2} \exp\left(-4\pi T_i \frac{h_{wi}(t_{i1})}{\tilde{Q}_i}\right) \tag{A3b}$$

where $\tilde{Q}_i$ is the average between flow rates $Q_i(t_{i1})$ and $Q_i(t_{i2})$ (according to Theis' assumption of constant flow rate) and $r_{we}$ is the effective radius [41] which integrates the skin factor $\sigma$ with expression $r_{we} = r_w e^{-\sigma}$. For this method, and for all the double-log methods presented in this work, if the logarithmic approximation does not hold, $T_i$ and $S_i$ can be inverted jointly from (A1) applied to aquifer $i$ at times $t_{i1}$ and $t_{i2}$ by using a numerical optimization method. However, as stated in [39], the logarithmic approximation is quickly reached in the pumping well, except in rare particular cases.

**Appendix B. Multi-Aquifer Model**

We present a semi-analytical solution for simulating transient drawdown in radially infinite confined multi-aquifers with homogeneous boundary conditions, wellbore storage, heterogeneous skin effects, and variable pumping flow rate. The presented model also accounts for a variable well radius, which is useful for deep wells in which the well radius decreases with depth. We consider $N_{aq}$ horizontal independent aquifers numbered from top to bottom (Figure A1) that are intercepted by the pumping well. When applying the pumping flow rate $Q_P$ to the well, mass balance in the well is expressed as:

$$Q_P + \sum_{i=1}^{N_{aq}} Q_i - Q_S = 0, \tag{A4}$$

where $Q_P$ is negative when the flow is extracted from the well, $Q_i$ is the (positive) flow rate pumped from aquifer $i$, and $Q_S$ is the (negative) wellbore storage flow rate. Note that $Q_P$ here is equivalent to $-Q$ in Appendix A.1.

The flow rates used in Equation (A4) are expressed as:

$$Q_S = S_w \frac{\partial h_{wS}}{\partial t}, \quad Q_i = 2\pi r_{wi} b_i K_i \frac{\partial h_i}{\partial r}\bigg|_{r=r_{wi}}, \tag{A5}$$

where $S_w$ is the well capacity defined as $S_w = \pi r_{wS}^2$ with $r_{wS}$ the wellbore storage radius (i.e., the radius of the zone where the water table fluctuates), $h_{wS}$ is the wellbore storage drawdown (i.e., the water surface level), $r_{wi}$ is the well radius at the depth of aquifer $i$, $b_i$ and $K_i$ are the conductive thickness and conductivity of aquifer $i$, respectively, and $h_i$ is the drawdown in aquifer $i$.

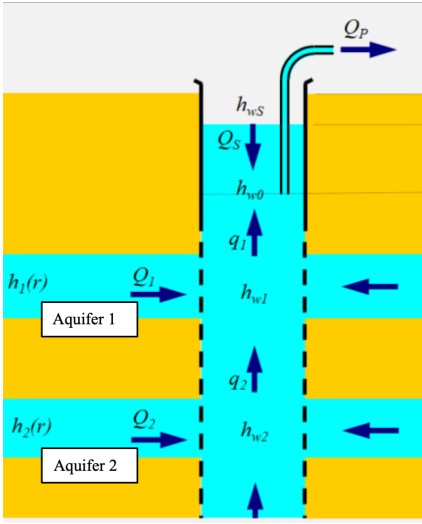

**Figure A1.** Schematic representation of the multi-aquifer system.

In each aquifer $i$ ($i = 1, \dots, N_{aq}$), the mass balance equation for a slightly compressible fluid is expressed as:

$$S_{si}\frac{\partial h_i}{\partial t} = \frac{K_i}{r}\frac{\partial}{\partial r}\left(r\frac{\partial h_i}{\partial r}\right) \tag{A6}$$

where $S_{si}$ is the specific storage of aquifer $i$, $r$ is the radial distance from the well center, and $t$ is the time elapsed since the pumping starts. Initial and boundary conditions are defined by:

$$h_i(r, t = 0) = 0, \quad \lim_{r \to +\infty} h_i(r, t) = 0, \tag{A7}$$

and the well skin effect is represented by a singular head loss as [42]:

$$h_{wi} = \left(h_i - r_{wi}\sigma_i\frac{\partial h_i}{\partial r}\right)_{|r=r_{wi}} \tag{A8}$$

with the dimensionless skin factor $\sigma_i$.

Applying the Laplace transform to Equation (A6) with the initial condition provided in (A7) leads to:

$$\frac{1}{\varepsilon_i}p\bar{h}_i = \frac{1}{r}\frac{\partial \bar{h}_i}{\partial r} + \frac{\partial^2 \bar{h}_i}{\partial r^2} \tag{A9}$$

where $p$ is the Laplace variable and $\varepsilon_i$ is the diffusivity defined as $\varepsilon_i = K_i/S_{si}$. Considering the boundary condition given in (A7), the solution for Equation (A9) is expressed as:

$$\bar{h}_i = C_{1i}\mathcal{I}_0(\gamma_i r) + C_{2i}\mathcal{K}_0(\gamma_i r) \tag{A10}$$

where $\mathcal{I}$ and $\mathcal{K}$ are the modified Bessel functions of the first and second kind, respectively, $\gamma_i = \sqrt{p/\varepsilon_i}$, and $C_{1i} = 0$ when applying the boundary condition from (A7). The derivative of $h_i$ is then expressed in the Laplace domain as:

$$\frac{\partial \bar{h}_i}{\partial r} = -C_{2i}\gamma_i\mathcal{K}_1(\gamma_i r). \tag{A11}$$

Expression (A8) is formulated in the Laplace domain as:

$$\bar{h}_{wi} = \left( \bar{h}_i - r_{wi}\sigma_i \frac{\partial \bar{h}_i}{\partial r} \right)_{|r=r_{wi}} \tag{A12}$$

and expressed as follows by using (A10) and (A11):

$$\bar{h}_{wi} = C_{2i}A_{1i}, \quad A_{1i} = \mathcal{K}_0(\gamma_i r_{wi}) + r_{wi}\sigma_i\gamma_i\mathcal{K}_1(\gamma_i r_{wi}). \tag{A13}$$

The following expressions of $h_i$ and its derivative in the Laplace domain are deduced from (A13):

$$\bar{h}_i = \bar{h}_{wi}\frac{\mathcal{K}_0(\gamma_i r)}{A_{1i}}, \qquad \frac{\partial \bar{h}_i}{\partial r}_{|r=r_{wi}} = -\bar{h}_{wi}\frac{\gamma_i\mathcal{K}_1(\gamma_i r_{wi})}{A_{1i}}. \tag{A14}$$

Expressing the Laplace transform of the vertical flow rate $Q_i$, defined in (A5), and combining it with the derivative expression in (A14) result in:

$$\bar{Q}_i = -A_{2i}\bar{h}_{wi}, \quad A_{2i} = 2\pi r_{wi}b_iK_i\gamma_i\mathcal{K}_1(\gamma_i r_{wi})/A_{1i}. \tag{A15}$$

Combining (A4), (A5) and (A15) result in:

$$S_w p \bar{h}_{wS} = \bar{Q}_P - \sum_{i=1}^{N_{aq}} A_{2i}\bar{h}_{wi}, \tag{A16}$$

and considering that the headloss in the well is negligible (i.e., $\bar{h}_{wi} = \bar{h}_{wS}$), $\bar{h}_{wS}$ is expressed as:

$$\bar{h}_{wS} = \frac{\bar{Q}_P}{S_w p + \sum_{i=1}^{N_{aq}} A_{2i}}. \tag{A17}$$

All the drawdowns and flow rates expressed in the Laplace domain are inverted in the time domain using [37]'s algorithm.

## Appendix C. Models for Transient Parameters and Properties

*Appendix C.1. Pumping Flow Rate Models*

We focus on expressions of flow rate models that have an explicit analytical formulation in the Laplace domain, for example exponential and polynomial formulations, as well as linear combinations of them. As the exponential decrease reproduces well the typical flow rate decrease that is observed due to the pump functioning [27], we consider this formulation with a decrease from flow rates $Q_{t_1}$ to $Q_{t_2}$ at times $t_1$ and $t_2$, respectively. This leads to the following pumping flow rate expression:

$$Q_P(t) = ae^{-t/b} + c, \tag{A18}$$
$$a = (Q_{t_1} - c)e^{t_1/b}, \quad c = (Q_{t_2} - Q_{t_1}\beta)/(1-\beta), \quad \beta = e^{(t_1-t_2)/b},$$

where $b$ is a fitting coefficient that controls the decrease shape.

The Laplace transform is expressed as:

$$\bar{Q}_P = a/(p + 1/b) + c/p \tag{A19}$$

considering that the pumping starts at time $t_1 = 0$.

*Appendix C.2. Couples of Equivalent Parameters $(S_{si}, \sigma_i)$*

For single-hole tests, the couples of parameters $(S_{si}, \sigma_i)$ and $(S'_{si}, \sigma'_i)$ are equivalent when they produce the same values of $h_{wS}$, $Q_S$ and $Q_i$. From the demonstration of the

multi-aquifer model provided in Appendix B, this is equivalent to produce the same values of $A_{2i}$ and $A'_{2i}$ that are defined as:

$$A_{2i} = \frac{2\pi r_{wi} b_i K_i \gamma_i \mathcal{K}_1(\gamma_i r_{wi})}{\mathcal{K}_0(\gamma_i r_{wi}) + r_{wi}\sigma_i \gamma_i \mathcal{K}_1(\gamma_i r_{wi})}, \tag{A20}$$

$$A'_{2i} = \frac{2\pi r_{wi} b_i K_i \gamma'_i \mathcal{K}_1(\gamma'_i r_{wi})}{\mathcal{K}_0(\gamma'_i r_{wi}) + r_{wi}\sigma'_i \gamma'_i \mathcal{K}_1(\gamma'_i r_{wi})}$$

where $\gamma'_i = \sqrt{pS'_{si}/K_i}$. Considering $A_{2i} = A'_{2i}$ results in:

$$\sigma'_i = \sigma_i - \frac{\mathcal{K}_0(\gamma'_i r_{wi})}{\gamma'_i r_{wi} \mathcal{K}_1(\gamma'_i r_{wi})} + \frac{\mathcal{K}_0(\gamma_i r_{wi})}{\gamma_i r_{wi} \mathcal{K}_1(\gamma_i r_{wi})}. \tag{A21}$$

Focusing on small arguments of the Bessel functions, the following approximations [43] are used:

$$\mathcal{K}_0(x) \approx -\ln(x/2) - \gamma_E, \quad \mathcal{K}_1(x) \approx 1/x, \tag{A22}$$

where $ln$ is the natural logarithm and $\gamma_E$ is the Euler constant. This leads to the following explicit relationship between the couples of properties $(S_{si}, \sigma_i)$ and $(S'_{si}, \sigma'_i)$:

$$S'_{si} = S_{si} \exp\left[2(\sigma'_i - \sigma_i)\right]. \tag{A23}$$

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
