# Peer review of "Models and Interpretation Methods for Single-Hole Flowmeter Experiments"

_water, doi:10.3390/w15162960_

Round 1

Reviewer 1 Report

1. Fig. 1 needs redrawing to be more clear and readable.

2. Fig.3 is like a table but categorized as a figure. I would suggest plotting a new figure that these data to be consistent with the explanation in the text.

3. I suggest separating the "discussion" and "conclusion" sections. Based on the content of the manuscript, the "discussion" section should be more comprehensive and impressive.

no comment

Reviewer 2 Report

Title : Models and interpretation methods for single-hole flowmeter experiments

 Authors: Gerard Lods and Delphine Roubinet

 Journal: WATER MDPI

MS Type: Research article

General comments

This study describe a new semi-analytical solution based on physics and that considers the main characteristics of the aquifers relaying on single-hole pumping test.

The authors presented two interpretation methods linked with single and multiple pumping tests, respectively: the Double Flowmeter Test with Transient Flowrate (DFTTF) and the Transient Flowrate Flowmeter Test (TFFT). Hence, they studied the impact of wellbore storage, transient pumping rate, and heterogeneous properties on the interpretation of synthetic data corresponding to single pumping tests on a double-aquifer system. They also analyzed the effect of the skin factor heterogeneity on transmissivity and storativity estimates through a kind of sensitivity analysis.

The paper is well written and presented. It is technically sound and of interest to the scientific community. I think it fits well with the themes of the Journal. Nevertheless, there is still a need for improvement, both to clarify the proposed methods and to qualify the significance (the interest) of the work, which has not been demonstrated in configurations that are more complex and based on real data.

Specific comments:

 Introduction

I understand that the study focuses on single well pumping tests and analytical approach to get aquifers properties; nevertheless, I think that your introduction should also mention inverse modeling approach. Perhaps the authors could refer to a study on interference well testing and the associated method which allows to identify groundwater flow parameters. See for instance:

Marinoni, M., Delay, F., Ackerer, P., Riva, M., & Guadagnini, A. (2016). Identification of groundwater flow parameters using reciprocal data from hydraulic interference tests. Journal of Hydrology, 539, 88-101.

Paragraph L65-70: You could add the article dealing with the interpretation of flowmeter data by Riva et al. (2012)

Riva, M., Ackerer, P., & Guadagnini, A. (2012). Interpretation of flowmeter data in heterogeneous layered aquifers. Journal of Hydrology, 452, 76-82.

I'm not a specialist in pumping tests, but I'm wondering about the existing literature on transient flows and the existing solutions for interpreting the measurements. It seems to me that the authors didn’t mention (at least in their introduction) other analytical - numerical approach to get the properties of the aquifer. See for instance:

Hemker, C. J. (1999). Transient well flow in vertically heterogeneous aquifers. Journal of Hydrology, 225(1-2), 1-18.

Kabala, Z. J., & El-Sayegh, H. K. (2002). Transient flowmeter test: semi-analytic crossflow model. Advances in water resources, 25(1), 103-121.

Zech, A., Müller, S., Mai, J., Heße, F., & Attinger, S. (2016). Extending Theis' solution: Using transient pumping tests to estimate parameters of aquifer heterogeneity. Water Resources Research, 52(8), 6156-6170.

Section 2.2

In equation (1b), the first time ti1 is selected to express Si. In fact, both measurement times ti1 and ti2 are contained in Ti. Since you are interested in transient flowrate, I suggest you add a short explanation just after equations 1.

 Section 2.3

I’m wondering if the two-aquifer system isn’t a specific case of the authors’ TFF test. I understand the idea of generalising the formulation of the problem and the writing of the equations.

Due to the different types of variable considered in your objective function, you should specify the definition of weights used in equation (2) (Line 336). I also suggest that the authors add references to the equations they used (and that are defined in appendix in the Laplace domain before inversion?) to calculate the flow and drawdown values given by the model (i.e. Qimodel(t) and hwsmodel(t)). How many times did you consider to calculate the objective function (for instance in section 3.3)?

 Section 3.2

Increasing differences between SFT / DFT and DTTF are observed when ɛij get smaller; the authors explain the homogeneous assumption of these methods prevents them from obtaining a correct result linked to a configuration with different storages and transmissivities between both aquifers.

For ɛij = 1, the homogeneous assumption is true and all the methods should be quite similar.

How do you explain that larger differences are observed when ɛij get around 103 (except for DFT method in configuration i) compared with larger values of ɛij?

I cannot observed the value of DFT method for ɛij = 10-6 in graphic 1i?

 Section 3.3

Notations Q1 and Q2 are perhaps a bit confusing because subscript can refer either to one of the aquifers or the time selection for transient flow.

Do you think that the properties of your two-aquifer system make it a truly heterogeneous system from a hydrogeological point of view? The authors should perhaps give orders of magnitude for the variables they consider (T, S, skin factor of real systems) so that the reader has a better understanding of their variability.

Discussion and conclusion

From a mathematical point of view, the proposed semi-analytical multi-aquifer model could allow to investigate a lot of effects (Lines 242-248). However, the authors should remind that they “only” tested it on a two-aquifer system with synthetic data.

Their interpretation methodology and primary sensitivity analysis on skin factor are also interesting but the potential of these developments hasn’t been fully demonstrated for more complex systems with more heterogeneous parameters and real data. This concluding part may require a few nuances.

 Appendix A1 (Line 313):

Molz et al. (1989) didn’t provide expression Ti=TQi/Q ? You combine their result (that considers a pseudo steady state condition) and your assumption (H1). It also corresponds to eq12 given by Kabala [22]. Is Q equivalent to Qp (according to previous notations in your article)?

Appendix A2 (Line 318)

Theoretical restrictions of the Theis’ solution are mentioned in [28] (p4).

Appendix B (Line 351)

Is a subscript “i” required in equation (A6) to characterize the radial distance from the well center which has been considered not constant?

Appendix C

Line 366: the boundary condition given in (A7), the solution for equation (A10A9) is expressed as

Line 430: “SsI’” and “SsI” in EqA23

A few typos

Line 4: I prefer flow rates (in two words); anyway, please harmonize the document and select either flowrate or flow rate.

Line 25 : planning

Line 26 : Heat pulse

Line 39 : vertical flowrates along over time

Line 41: to wait until that the system

Line 62: due to the pump functioning,

Line 66: …heterogeneities of the considered systems, considering assuming for instance homogeneous hydraulic diffusivities and storativities in the system.

Line 72 & 272 : analyse

Line 107: conductive

Line 122: to interpret multiple-pumping local-log experiments

Line 129: model the data

Line 150: with the TFT method

Line 152: the wellbore storage effect

Line 153: A homogeneous pumping well of radius 8 cm

Line 185: and thus tending to verify the homogeneous diffusivity assumption

Line 192: the ratio of estimated to the true value

Line 233: to estimate an equivalent couple

Line 252: these estimates in relation with to the skin factor

Line 261: while the storativity is usually estimated from data collected in an observation borehole

Line 264: we will consider structural information that helps to reduce the range

I'm not a native English speaker but the manuscript seems well written.

Round 2

Reviewer 1 Report

The authors fully respond to my comments and suggestions. 

Reviewer 2 Report

The authors improved their manuscript and addressed all my comments.

I think that the method is now explained more clearly and that the limitations of their approach are explicitly mentioned, as well as the perspectives relating in particular to an application on real cases.

Regards